# The rates and measurement of adherence to acamprosate in randomised controlled clinical trials: A systematic review

**Kim Donoghue**[1,2]ᴓ*, **Laura Hermann**[2,3]ᴓ, **Eileen Brobbin**[2]ᴓ, **Colin Drummond**[2]ᴓ

**1** Department of Clinical, Educational and Health Psychology, University College London, London, United Kingdom, **2** National Addictions Centre, Addictions Department, Institute of Psychiatry, Psychology and Neuroscience, King's College London, London, United Kingdom, **3** Institute of Clinical and Applied Health Research, University of Hull, Hull, United Kingdom

ᴓ These authors contributed equally to this work.
* kim.donoghue@ucl.ac.uk

## Abstract

### Aim

The current research aims to systematically review the rates of adherence reported in randomised controlled clinical trials of acamprosate. It also sought to determine the reliability of the adherence monitoring and measurement methods used in these trials.

### Methods

The protocol for this review was pre-registered (PROSPERO: CRD42021230011). A search of the literature was conducted using OVID MEDLINE, Embase and PsycINFO from database inception to January 2021. Randomised controlled trials with a minimum sample size of 10 per treatment arm that compared the efficacy of acamprosate with placebo or other active medication in adults with a diagnosis of alcohol dependence were included. Data on rates of adherence, methods of measurement and monitoring of adherence was extracted from eligible studies independently in duplicate by two reviewers. A weighted mean adherence rate was calculated. The reliability of adherence monitoring methods was determined by calculating an adherence-assurance score based on the adherence monitoring method used. Risk of bias was assessed using the Cochrane Risk of Bias Tool.

### Results

Fifteen studies met the eligibility criteria involving 4,450 participants (2,480 participants in the placebo arms). A mean adherence rate of 88% (54.2–95.0%) was reported across studies that reported the percentage of medication taken. A mean adherence rate of 84.9% (56.4–91.3%) was reported for trials that reported the percentage of participants taking more than 80% of medication prescribed. There is low confidence in the methods used to monitor adherence with all clinical trials having a low adherence-assurance rating. Risk of bias was judged to be high for all included studies.

**Data Availability Statement:** All relevant data are within the paper and its Supporting Information files.

**Funding:** CD is part funded by the NIHR Biomedical Research Centre at South London and Maudsley NHS Foundation Trust and King's College London, and by the NIHR Collaboration for Leadership in Applied Health Research and Care South London (NIHR CLAHRC South London) now recommissioned as NIHR Applied Research Collaboration South London, and receives funding from an NIHR Senior Investigator award. The funders had no contribution to the study design; in the collection, analysis, and interpretation of data; in the writing of the report; and in the decision to submit the article for publication. The views expressed are those of the authors and not necessarily those of the National Health Service (NHS), the NIHR or the Department of Health and Social Care.

**Competing interests:** The authors have declared that no competing interests exist.

## Conclusions

Adherence to acamprosate in clinical trials can be poor with low confidence in the methods used to measure it. Adherence rates therefore might not be accurate, which has implications for determining the efficacy of acamprosate.

## Introduction

Alcohol consumption is a leading factor for disease burden worldwide, associated with 60 acute and chronic health conditions and the leading cause of premature death in those aged 15–49 years. In 2016, alcohol consumption was attributable to 2.8 million deaths worldwide [1]. Those requiring treatment for their alcohol use often undergo frequent episodes of withdrawal and resumption of drinking with up to 70% of people returning to drinking in the year following treatment [2].

Acamprosate is a safe, effective and cost-effective medication to help support relapse prevention [3]. Guidelines produced by the National Institute for Health and Care Excellence (NICE) recommend acamprosate as a first-line treatment, in conjunction with psychosocial therapy, to help support those who have completed alcohol withdrawal to remain alcohol free [4]. Acamprosate modulates the glutamatergic system and stabilises the imbalance between inhibitory (GABA) and excitatory (glutamate) neurotransmitters in the brain during alcohol withdrawal, whereby reducing the conditioned effect of alcohol and the negative reinforcement of the addiction [5–7].

Despite the therapeutic potential of acamprosate, poor adherence to the medication poses a problem for effectiveness in clinical practice. Adherence to a medication can be considered the extent to which a patient's actions match the recommendations agreed with the prescriber [8]. Suboptimal outcomes may result from underdosing, overdosing or taking medication at incorrect intervals. Improved treatment outcomes for alcohol dependence are associated with better adherence to medications for alcohol relapse prevention [9, 10]. Medication adherence is a common problem across clinical care but is particularly an issue in chronic conditions and greater risk of poor adherence has been associated with those who misuse substances [11]. Since clinical trials offer a controlled environment where adherence can be monitored by research staff and payment may even be received for participation, medication non-adherence in clinical practice is likely to be substantially greater than in clinical trials.

The precise measurement of adherence in clinical trials is essential to accurately assess the efficacy of the medication under investigation. Methods for monitoring adherence in clinical trials include direct supervision, pill count, patient or clinician self-report, biochemical markers and electronic adherence monitoring. Pill count and patient self-report are often used to measure adherence in clinical trials, they can be inexpensive and place minimal burden on the participant. However, self-report may lead to an over-estimate of rates of adherence [12]. Electronic adherence monitoring that involves a medication bottle cap (e.g. Medication Events Monitoring System) that records when the bottle is opened is considered the gold-standard for clinical trials but is not feasible for routine clinical care [13].

The current research aims to systematically review the rates of adherence reported in randomised controlled clinical trials of acamprosate. It also sought to determine the reliability of the adherence monitoring and measurement methods used in these trials.

## Materials and method

The protocol for this systematic review was preregistered with the International Prospective Register of Systematic Reviews (PROSPERO: CRD42021230011). This paper complies with the PRISMA (Preferred Reporting Items for Systematic Reviews and Meta-analyses).

### Eligibility criteria

Studies were included if they were randomised controlled trials with a minimum sample size of 10 per treatment arm comparing the efficacy of acamprosate with placebo or other active medication for alcohol relapse prevention. Included studies were of adults (aged 18 or older) with a diagnosis of alcohol dependence (ICD or DSM). Studies were excluded if they used a cross-over or open label design or included pregnant women. The eligibility of trials was confirmed in line with the inclusion/exclusion criteria.

### Information source and search strategy

The electronic databases EMBASE, MEDLINE and PsycINFO (using the Ovid interface) were searched from database inception to the 3rd January 2021, combing terms for alcohol dependence, acamprosate and randomised controlled trials (see S1 File for the full search strategy). Searches were limited to studies published in English.

### Study selection and data collection process

Search results were managed using Endnote and Microsoft Excel. Two reviewers (K.D. and L. H.) independently screened the titles and abstracts of all identified references. The full texts of potentially relevant articles were then screened independently in duplicate by the two reviewers. Any disagreement when screening titles, abstracts or full text documents was resolved by discussion between the two reviewers. Data from each relevant article was extracted independently in duplicate by the two reviewers using Microsoft Excel spreadsheets that had been pre-piloted. Discrepancies were discussed between the two reviewers and agreement reached. The Data extracted included; participant characteristics (number of participants in each study arm, age, gender, ethnicity), study characteristics (number of trial arms, length of treatment with acamprosate and comparator/placebo, country of the study, psychosocial intervention), medication adherence monitoring method (e.g. pill count), the frequency that adherence was measured (days), adherence rate (acamprosate and comparator/placebo), the measure of medication adherence used (e.g. % of prescribed medication taken), overall adherence rate (acamprosate and comparator/placebo).

### Outcome measures

The following outcomes were assessed; 1. Medication adherence rate, 2. Medication adherence monitoring method, 3. Frequency of adherence measurement (percentage of days that the monitoring was used), 4. Measure of medication adherence, 5. Length of treatment with medication (days).

### Data synthesis

Adherence rates for each trial was combined and weighted by sample size according to the adherence reporting method used. Where separate percentages were reported for active medication and placebo groups, the percentage of medication for the active medication group was taken. If separate percentages are reported for different sub-groups, for example type of psychological therapy received, all were included in the adherence rate calculations that were

completed in Microsoft Excel. There was variation in the treatment length for the clinical trials included. Pearson's correlation using IBM SPSS version 26 [14] was used to conduct a post hoc exploration of the impact of length of treatment on rate of adherence.

The method described by Swift et al. [15] was used to calculate an adherence-assurance score for the trials included in this review. All data was entered into Microsoft Word and calculated manually. To calculate the adherence-assurance score, adherence monitoring methods were assigned a monitoring confidence level which takes into consideration ability for circumvention; high/3 = supervision, medium/2 = Medication Events Monitoring System (MEMS), riboflavin or acamprosate levels, Low/1 = self-report, pill count, blister packs. The percentage of dosing days on which the monitoring method was used was calculated. Biological testing methods such as the presence of riboflavin were considered to provide confirmation of dosing on a single day. An adherence-assurance score was then calculated using the following formula; Adherence-assurance score = (monitoring confidence level) X (monitoring frequency).

Where multiple methods of adherence monitoring had been used concurrently a combined score was calculated by adding together the two adherence-assurance scores for the two methods. For trials that used two methods with the same confidence level (e.g. two low confidence methods such as pill count and self-report), no additional adherence assurance was allocated. Raw adherence-assurance scores were normalised to 100% and assigned an adherence-assurance rating of low (0–49%), medium (50–79%) or high (80–100%).

## Risk of bias

Eligible studies were assessed for risk of bias using the Cochrane Risk of Bias Tool [16], which included risk of bias arising from the randomisation process, deviations from the intended interventions, missing outcome data, measurement of the outcome and selection of the reported results Two reviewers assessed each relevant article independently with discrepancies resolved by discussion.

## Results

Fig 1 shows the results of the systematic search of the literature. A total of 15 studies were included involving 4,450 participants (2,480 participants in the placebo arms), the characteristics of these studies are presented in Table 1. The length of treatment with acamprosate ranged from 84 days to 365 days.

Nine studies reported the percentage of prescribed acamprosate taken with a weighted mean of 81.5% (range; 54.2% to 95%) [17–25]. Pearson's correlation found no statistically significant correlation between the length of treatment with acamprosate and the percentage of prescribed medication taken (r = -0.308, p = 0.357). Two studies reported the proportion of participants taking at least 80% of prescribed acamprosate with a weighted mean of 84.9% of participants (range; 56.4% to 91.3%) [26, 27]. Three studies did not report adherence rates but stated that there were no differences between the acamprosate and placebo groups [28–30]. The final trial reported 76.9–84.5% of participants had regular intake of acamprosate during the trial with no differences between groups but no definition of regular intake was given [31]. The adherence assurance scores are reported in Table 2, all of which had a low adherence-assurance rating. All but one trial [29] used pill count to assess adherence to acamprosate, with ten of the trials relying solely on pill count. In addition to pill count, two trials utilised biological methods, Mason et al., [25] used plasma to monitor acamprosate adherence and Sass et al., [30] used urine analysis of acamprosate. Investigator assessment was used in addition to pill count by one trial [31] and a daily monitoring card was used by another trial [26]. A daily dosing card completed by participants was used as the only method of adherence assessment in in

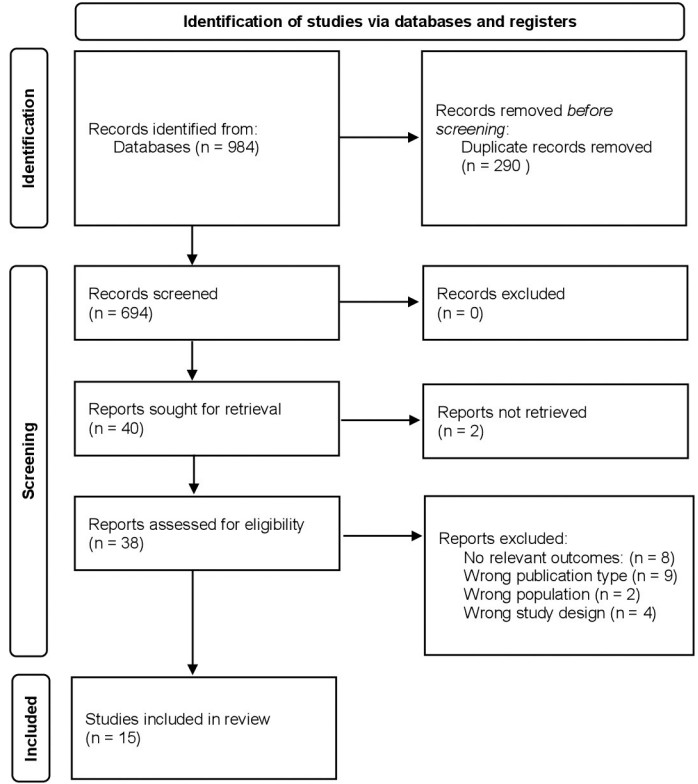

**Fig 1. PRISMA flow diagram.**

one trial [29]. All eligible studies were judged to have a high risk of bias, which was largely due to risk of bias arising from missing data (S1 Table).

## Discussion

Percentage of acamprosate taken during the clinical trials varied from 54.2% of prescribed medication taken to as high as 95%. However, the reliability of the methods used to measure adherence is low with the majority of trials relying on pill count. The risk of bias for the included trials was high, this was largely due to risk of bias arising from missing data. Dropout rates for the included trials was often high and it was unclear if this was taken into consideration when calculating adherence. Adherence rates may therefore have been inflated by only including those who completed the trial, which would be biased towards those who were adherent to acamprosate.

Since clinical trials offer a controlled environment where adherence can be monitored by research staff, medication non-adherence in usual clinical practice is likely to be greater. NICE [4] recommends that pharmacotherapies for alcohol relapse prevention are taken for at least 6 months, however, the length of time that these medications are taken often falls short of this. [32]. Therefore, service users may not be gaining the maximum benefit from acamprosate through poor medication adherence and not taking the medication for a sufficient period of time.

The impact of medication adherence on treatment effectiveness has been explored in research investigating the effect of adherence to acamprosate on alcohol outcomes. There is some evidence to suggest that non-adherence to acamprosate early in treatment is associated

**Table 1. Study characteristics.**

| Study | Country | Total N Acamprosate | Total N placebo | Participant age, mean (SD) | Participant gender, % male | Measure of adherence | Adherence rate Acamprosate | Adherence rate Placebo | Treatment length (days) | Risk of bias |
|---|---|---|---|---|---|---|---|---|---|---|
| Anton (2006) | USA | 302 (MM = 152 and CBI = 150) | 308 (MM = 153 and CBI = 156) | MM Acamprosate: 44.0 (SD 10.97 Placebo: 44.2 (SD 9.15) CBI Acamprosate: 45.4 (SD 10.32) Placebo: 43.2 (SD 9.74) | MM Acamprosate: 69.1% Placebo: 67.3% CBI Acamprosate: 70.9% Placebo: 70.5% | % prescribed meds taken | 84.2% | NR | 112 | High |
| Berger (2013) | USA | 51 | 49 | Acamprosate: 46.6 (7.7) Placebo: 47.7 (8.5) | Acamprosate: 58.8% Placebo: 65.3% | % of prescribed meds taken | 93.3% | 91.6% | 84 | Some concerns |
| Besson (1998) | Switzerland | 55 | 55 | Acamprosate: 42.7 (NR) Placebo: 42.2 (NR) | Acamprosate: 83.6% Placebo: 76.4% | Not reported | Not reported but non-significant difference between groups noted except on the last study visit–those on placebo took significantly fewer tablets | Not reported but non-significant difference between groups noted except on the last study visit–those on placebo took significantly fewer tablets | 360 | High |
| Geerlings (1997) | Benelux i.e. The Netherlands, Belgium and Luxembourg | 128 | 134 | Acamprosate: 40.3 (9.2) Placebo: 41.7 (8.1) | Acamprosate: 76% Placebo: 76% | % medication taken | 86% | 88% | 180 | High |
| Gual (2001) | Spain | 141 | 147 | Acamprosate: 41.4 (9.01) Placebo: 40.7 (9.47) | Acamprosate: 80% Placebo: 79% | Average % medication taken per day | 91.5% | 97.8% | 180 | High |
| Higuchi (2015) | Japan | 163 | 164 | Acamprosate: 51.7 (12.4) Placebo: 53.1 (12.2) | Acamprosate: 86.5% Placebo: 88.4% | Not reported | Both groups included 7 pts whose adherence rate was below 70%, no intergroup differences | Both groups included 7 pts whose adherence rate was below 70%, no intergroup differences | 168 | High |
| Kiefer (2003) | Germany | 40 | 40 | Acamprosate: 46.3 (7.7) Placebo: 45.6 (11.1) | Acamprosate: 75% Placebo: 68% | % prescribed medication taken | 81.1% overall group specific not reported (Non-significant difference between groups noted | 81.1% overall group specific not reported (Non-significant difference between groups noted | 84 | High |
| Mann (2013) | Germany | 172 | 86 | Acamprosate: 45.1 (8.5) Placebo: 46.6 (9.3) | Acamprosate: 80% Placebo: 78% | % prescribed medication | 76.7% | 73.5% | 84 | High |
| Mason (2006) | USA | 2g/day = 258 3g/day = 83 | 260 | Acamprosate: 2g/ day = 44.9 (10.5), Acamprosate: 3g/ day = 43.6 (8.9). Placebo = 44.5 (10.0) | Aamprosate 2g/ day = 70% Acamprosate 3g/ day = 71% Placebo = 64% | % prescribed medication | 2g/day = 89.0%. 3g/day = 88.5% | 92.6% | 168 | High |
| Morley (2006) | Australia | 55 | 61 | Acamprosate: 45.2 (9.2) Placebo: 42.4 (9.3) | Acamprosate: 76.4% Placebo: 71.7% | % taking 80% medication | 56.4% | 50.8% | 84 | High |
| Paille (1995) | France | 361 (Dose 1.3g/day = 188 and Dose 2g/day = 173) | 177 | Acamprosate 1.3g/day: 43.7 (8.6) Acamprosate 2g/day: 43.3 (9.3) Placebo: 42.5 (8.9) | Acamprosate 1.3g/ day: 77.7% Acamprosate 2g/day: 79.2% Placebo: 83.1% | % pills taken | **90 days:** Acp 1.3g/ day = 80.3%, Acp 2g/day = 76.8%. **180 days:** ACP 1.3g/ day = 70.5%, ACP 2g/day = 54.2%. **360 days:** ACP 1.3g/ day = 54.9%, ACP 2g/day = 35.8% | **90 days** = 81.9%. **180 days:** 64.4%. **360 days** = 47.3% | 365 | High |
| Pelc (1997) | Belgium and France | 126 (1332mg/day = 63 and 1998mg/day = 63) | 62 | NR | NR | % pills taken | 95% of tablets not returned—overall, no group specific values reported | 95% of tablets not returned—overall, no group specific values reported | 90 | High |
| Sass (1996) | Germany | 136 | 136 | Acamprosate: 41.9 (8.4) Placebo: 40.5 (8.6) | Acamprosate: 75% Placebo: 80% | Not reported | Not reported but no difference between groups noted | Not reported but no difference between groups noted | 336 | High |
| Tempesta (2000) | Italy | 164 | 166 | Acamprosate: 45.9 (11.33) Placebo: 45.9 (11.19) | Acamprosate: 84.8% Placebo: 80.7 | Regular intake of study medication | 76.9% - 84.5% 'regular intake', no group specific values reported, non-significant difference between groups noted | 76.9% - 84.5% 'regular intake', no group specific values reported, non-significant difference between groups noted | 180 | High |
| Wolwer (2011) | Germany | IBT: 124 TAU: 122 | 125 | Acamprosate IBT: 45.1 (7.8) Acamprosate TAU: 45.7 (7.5) | Acamprosate IBT: 75.8% Acamprosate TAU: 71.3% Placebo: 62.2% | % participants taking 80% or more of medication | 91.3% overall | 91.3% overall | 182 | High |

**Table 2. Adherence-assurance rating for acamprosate.**

| Study | Method | Monitoring method A | | | Monitoring method B | | | | Raw score (%) | Normal score (%) | Adherence assurance rating |
|---|---|---|---|---|---|---|---|---|---|---|---|
| | | Adherence assurance score | Frequency (%) | Subscore | Method/ confidence | Adherence assurance score | Frequency (%) | Subscore | | | |
| **Acamprosate** | | | | | | | | | | | |
| Anton 2006 | Pill count | 1 | 100 | 100 | | | | | 100 | 33 | Low |
| Berger 2013 | Pill count | 1 | 100 | 100 | | | | | 100 | 33 | Low |
| Besson 1998 | Pill count | 1 | 100 | 100 | | | | | 100 | 33 | Low |
| Geerlings 1997 | Pill count | 1 | 100 | 100 | | | | | 100 | 33 | Low |
| Gual 2001 | Not reported | | | | | | | | | | |
| Higuchi 2015 | Self-complete daily dosing diary | 1 | 100 | 100 | | | | | 100 | 33 | Low |
| Kiefer 2003 | Pill count | 1 | 100 | 100 | | | | | 100 | 33 | Low |
| Mann 2013 | Pill count | 1 | 100 | 100 | | | | | 100 | 33 | Low |
| Mason 2006 | Pill count | 1 | 100 | 100 | Plasma acamprosate | 2 | 2 | 4 | 104 | 35 | Low |
| Morley 2006 | PIll count and self report | 1 | 100 | 100 | Daily monitoring card | 1 | 100 | 100 | 100 | 33 | Low |
| Paille 1995 | Pill count | 1 | 100 | 100 | | | | | 100 | 33 | Low |
| Pelc 1997 | Pill count | 1 | 100 | 100 | | | | | 100 | 33 | Low |
| Sass 1996 | Pill count | 1 | 100 | 100 | Urine-analysis of acamprosate levels | 2 | Not reported | Unknown | 100 | 33 | Low |
| Tempesta 2000 | Pill count | 1 | 100 | 100 | Investigator assessment | 1 | 100 | 100 | 100 | 33 | Low |
| Wölwer 2011 | Pill count | 1 | 100 | 100 | | | | | 100 | 33 | Low |

with poorer drinking outcomes [9, 33]. Effective methods of improving adherence to medications for alcohol relapse prevention are needed in both clinical trials and clinical practice. Simple interventions such as using text messages, dosette boxes and alarms to remind patients to take their medication are of value [34–36]. More complex psychosocial interventions such as Compliance Enhancement Therapy (CET) and Medical Management (MM) have been successfully used in clinical trials to support improved adherence to medication for those with alcohol dependence by promoting positive beliefs about medication and patient addressing concerns [17, 37]. Despite the successful inclusion of psychosocial interventions to enhance adherence in clinical trials, there has been little research into its application in a more typical clinical setting. Psychosocial interventions supporting adherence to medications for alcohol relapse prevention may not be directly transferable to clinical practice due to the burden on staff and costs of delivery [38]. Further research into how we can best support people completing treatment for alcohol dependence to take acamprosate as prescribed is needed.

This systematic review has identified a low confidence in the measures used to report adherence. A hierarchy from low to high confidence in the method used to monitor adherence

to naltrexone has been proposed by Swift et al. [15]. The hierarchy, based on a patient's ability to evade measurement of adherence, considered patient self-report and counting returned pills to have a "low" confidence. A "medium" confidence was assigned to Medication Events Monitoring System (MEMS) caps to electronically monitor pill bottle opening, or biomarkers such as the addition of riboflavin. Methods considered to have a "High" confidence included supervision of dosing, long-acting injectable preparations, or monitoring of the level of prescribed medication in the blood. Poor measurement and reporting of adherence to medications in clinical trials may lead to incorrect assertions about efficacy being made. Robust measurement is essential to ensure an accurate picture of medication adherence, which may be achieved using a combination of methods that are high/medium as well as low confidence. The implementation of standardised reporting of adherence rates and accurate, high confidence adherence measurement methods in clinical trials would assist with comparison of efficacy results across trials.

## Limitations

The results of this systematic review should be interpreted considering some limitations to the research. Studies were heterogeneous in their method, measurement and reporting of adherence to acamprosate making comparison between studies difficult and further subgroup analysis not possible. Adherence measures and methods were often poorly described in the trial papers making it difficult to determine the impact of missing data bias. The review only included papers published in English due to the language limitations of the authors. We were unable to assess publication bias in this systematic review. It is possible that publication bias could have led to an overestimation of the rates of adherence to acamprosate in clinical trials. There is an association between adherence to acamprosate and its efficacy and therefore unpublished trials with negative results may have had greater rates of non-adherence to acamprosate.

## Conclusions

The efficacy of acamprosate for alcohol relapse prevention is well documented. However, poor adherence to acamprosate may impact on its effectiveness in clinical practice. Pill count was the most common method of monitoring adherence, which has a low confidence. The method for measuring adherence was often poorly described and varied across studies identified in this review; harmonisation of these methods across studies would make comparison easier and results more transparent.

## Supporting information

**S1 Checklist. PRISMA 2020 for abstracts checklist.**
(DOCX)

**S2 Checklist. PRISMA 2020 checklist.**
(DOCX)

**S1 File. Search strategy.**
(PDF)

**S1 Table. Risk of bias.**
(DOCX)

## Author Contributions

**Conceptualization:** Kim Donoghue, Laura Hermann, Eileen Brobbin, Colin Drummond.

**Data curation:** Kim Donoghue, Laura Hermann.

**Formal analysis:** Kim Donoghue, Laura Hermann.

**Investigation:** Laura Hermann.

**Methodology:** Kim Donoghue.

**Project administration:** Kim Donoghue.

**Supervision:** Kim Donoghue, Colin Drummond.

**Writing – original draft:** Kim Donoghue.

**Writing – review & editing:** Kim Donoghue, Laura Hermann, Eileen Brobbin, Colin Drummond.

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
