## [Editor Report · Decision Letter 0]

14 Nov 2021

PONE-D-21-32169The rates and measurement of adherence to acamprosate in randomised controlled clinical trials: A systematic reviewPLOS ONE

Dear Dr. Donoghue,

Thank you for submitting your manuscript to PLOS ONE. After careful consideration, we feel that it has merit but does not fully meet PLOS ONE’s publication criteria as it currently stands. Therefore, we invite you to submit a revised version of the manuscript that addresses the points raised during the review process.

We look forward to receiving your revised manuscript.

Kind regards,

Tariq Jamal Siddiqi

Academic Editor

PLOS ONE

Journal Requirements:

2. Thank you for submitting the above manuscript to PLOS ONE. During our internal evaluation of the manuscript, we found significant text overlap between your submission and the following previously published works, some of which you are an author.

- http://docplayer.net/13017100-Protocol-number-page-1-of-30.html

The text that needs to be addressed involves the final paragraph of the Discussion.

Please revise the manuscript to rephrase the duplicated text. Please note that further consideration is dependent on the submission of a manuscript that addresses these concerns about the overlap in text with published work.

Additional Editor Comments:

Donoghue et al. has performed a systematic review on, “The rates and measurement of adherence to acamprosate in randomised controlled clinical trials” in which they show that adherence to acamprosate is poor. In my opinion, this study can be improved by incorporating the following points:

1. The exclusion criteria needs to be clearly mentioned in the methods section of the study.

2. Mentioning which particular factors were assessed in the risk of bias of RCTs can improve the quality of methods.

3. The company for the software used in data analysis is not mentioned.

4. Any sensitivity analysis that was performed is not clearly mentioned in the study.

5. The results can be improved by including the mentioning the results of trial quality assessment as well.

6. There is no evidence for how the heterogeneity of the results was improved.

7. It is not stated whether any tests for assessing the publication bias were used or not.

8. Future implications regarding the trials required or the gap in literature which can be covered is not clearly mentioned.

9. Subgroups analysis was not performed on the results which would have improved the quality of results further.

10. The discussion can be improved by discussing more about the gaps in the literature and how this study has identified new aspects in this field which can be significant in improving the medical literature.
---

## [Author Response · Author response to Decision Letter 0]

15 Dec 2021

Journal requirements

1. Please ensure that your manuscript meets PLOS ONE's style requirements, including those for file naming

Thank you for raising this, we have consulted PLOS ONE’s style requirements and amended as applicable. 

2. Thank you for submitting the above manuscript to PLOS ONE. During our internal evaluation of the manuscript, we found significant text overlap between your submission and the following previously published works, some of which you are an author.

Apologies for this oversight. The work in question is our unpublished trial protocol that is freely available through the funder’s website. The text has been amended as below;

Original text

Swift et al. [15] suggest a hierarchy from low to high confidence in the method of adherence monitoring for naltrexone based on a patient’s ability to evade measurement of adherence. Patient self-report, counting of returned medication or inspection of blister packs were assigned “low” confidence, electronic monitoring of pill bottle opening (Medication Events Monitoring System (MEMS) caps) or biomarkers such as the addition of riboflavin were assigned a “medium” confidence. “High” confidence was assigned to supervised dosing, long-acting injectable preparations, or monitoring of blood levels of the prescribed medication.

Updated text

A hierarchy from low to high confidence in the method used to monitor adherence to naltrexone has been proposed by Swift et al. [15]. The hierarchy, based on a patient’s ability to evade measurement of adherence, considered patient self-report and counting returned pills to have a “low” confidence. A “medium” confidence was assigned to Medication Events Monitoring System (MEMS) caps to electronically monitor pill bottle opening, or biomarkers such as the addition of riboflavin. Methods considered to have a “High” confidence included supervision of dosing, long-acting injectable preparations, or monitoring of the level of prescribed medication in the blood. 

Additional Editor Comments

1. The exclusion criteria needs to be clearly mentioned in the methods section of the study.

The exclusion criteria have been stated more clearly in the methods as follows (line 100-101);

Studies were excluded if they used a cross-over or open label design or included pregnant women.

2. Mentioning which particular factors were assessed in the risk of bias of RCTs can improve the quality of methods.

The following text has been added (line 155-157);

Eligible studies were assessed for risk of bias using the Cochrane Risk of Bias Tool [16], which included risk of bias arising from the randomisation process, deviations from the intended interventions, missing outcome data, measurement of the outcome and selection of the reported results.

3. The company for the software used in data analysis is not mentioned.

The following has been added (line 133);

If separate percentages are reported for different sub-groups, for example type of psychological therapy received, all were included in the adherence rate calculations that were completed in Microsoft Excel.

And also (line 138-139)

The method described by Swift et al. [15] was used to calculate an adherence-assurance score for the trials included in this review. All data was entered into Microsoft Word and calculated manually.

4. Any sensitivity analysis that was performed is not clearly mentioned in the study.

We explored the relationship between study length and adherence and found no relationship (line 169 to 171);

Pearson’s correlation found no statistically significant correlation between the length of treatment with acamprosate and the percentage of prescribed medication taken (r=-0.308, p=0.357). 

5. The results can be improved by including the mentioning the results of trial quality assessment as well.

Risk of bias is already mentioned on (line 183-184);

All eligible studies were judged to have a high risk of bias, which was largely due to risk of bias arising from missing data (S2 table).

6. There is no evidence for how the heterogeneity of the results was improved.

This paper sought to determine the rates of adherence to acamprosate and the reliability of the adherence monitoring and measurement methods. Identifying heterogeneity was therefore part of the aims of the paper. We therefore do not think it is appropriate to look at improving heterogeneity for this paper. 

7. It is not stated whether any tests for assessing the publication bias were used or not.

Thank you for highlighting this, we have added the following to the limitations (line 242-246);

We were unable to assess publication bias in this systematic review. It is possible that publication bias could have led to an overestimation of the rates of adherence to acamprosate in clinical trials. There is an association between adherence to acamprosate and its efficacy and therefore unpublished trials with negative results may have had greater rates of non-adherence to acamprosate.

8. Future implications regarding the trials required or the gap in literature which can be covered is not clearly mentioned.

Thank you for this suggestion we have added the following text (line 215-216);

Despite the successful inclusion of psychosocial interventions to enhance adherence in clinical trials, there has been little research into its application in a more typical clinical setting. Psychosocial interventions supporting adherence to medications for alcohol relapse prevention may not be directly transferable to clinical practice due to the burden on staff and costs of delivery [17, 37]. Further research into how we can best support people completing treatment for alcohol dependence to take acamprosate as prescribed is needed. 

9. Subgroups analysis was not performed on the results which would have improved the quality of results further.

Thank you for this suggestion. There was a wide variation in the reporting of adherence with some studies using percentage of participants who took 80% of medication prescribed and others reporting the mean percentage of medication taken. Consequently, there was limited data to perform further subgroup analysis. The following has been added to the limitations section (line 239-240);

Studies were heterogeneous in their method, measurement and reporting of adherence to acamprosate making comparison between studies difficult and further subgroup analysis not possible.

10. The discussion can be improved by discussing more about the gaps in the literature and how this study has identified new aspects in this field which can be significant in improving the medical literature.

Please see number 8 above and we have also amended the following to be clearer in our discussion about the gaps in current research (line 233-234);

The implementation of standardised reporting of adherence rates and accurate, high confidence adherence measurement methods in clinical trials would assist with comparison of efficacy results across trials.

---

## [Decision Letter · Decision Letter 1]

18 Jan 2022

The rates and measurement of adherence to acamprosate in randomised controlled clinical trials: A systematic review

PONE-D-21-32169R1

Dear Dr. Donoghue,

We’re pleased to inform you that your manuscript has been judged scientifically suitable for publication and will be formally accepted for publication once it meets all outstanding technical requirements.

Kind regards,

Tariq Jamal Siddiqi

Academic Editor

PLOS ONE

Additional Editor Comments (optional):

Reviewers' comments:

Reviewer's Responses to Questions

**Comments to the Author**

1. If the authors have adequately addressed your comments raised in a previous round of review and you feel that this manuscript is now acceptable for publication, you may indicate that here to bypass the “Comments to the Author” section, enter your conflict of interest statement in the “Confidential to Editor” section, and submit your "Accept" recommendation.

Reviewer #1: All comments have been addressed

Reviewer #2: All comments have been addressed

2. Is the manuscript technically sound, and do the data support the conclusions?

Reviewer #1: Yes

Reviewer #2: Yes

3. Has the statistical analysis been performed appropriately and rigorously? 

Reviewer #1: Yes

Reviewer #2: Yes

4. Have the authors made all data underlying the findings in their manuscript fully available?

Reviewer #1: Yes

Reviewer #2: Yes

5. Is the manuscript presented in an intelligible fashion and written in standard English?

Reviewer #1: Yes

Reviewer #2: Yes

6. Review Comments to the Author

Reviewer #1: (No Response)

Reviewer #2: (No Response)

7. PLOS authors have the option to publish the peer review history of their article (what does this mean?). If published, this will include your full peer review and any attached files.

Reviewer #1: No

Reviewer #2: No

---

## [Editor Report · Acceptance letter]

24 Jan 2022

PONE-D-21-32169R1 

The rates and measurement of adherence to acamprosate in randomised controlled clinical trials: A systematic review 

Dear Dr. Donoghue:

I'm pleased to inform you that your manuscript has been deemed suitable for publication in PLOS ONE. Congratulations! Your manuscript is now with our production department. 

Kind regards, 

on behalf of

Dr. Tariq Jamal Siddiqi 

Academic Editor

PLOS ONE